# Bacterial Persister-Cells and Spores in the Food Chain: Their Potential Inactivation by Antimicrobial Peptides (AMPs)

**DOI:** 10.3390/ijms21238967

**Published:** 2020-11-27

**Authors:** Shiqi Liu, Stanley Brul, Sebastian A. J. Zaat

**Affiliations:** 1Swammerdam Institute for Life Sciences, Department of Molecular Biology and Microbial Food Safety, University of Amsterdam, 1098 XH Amsterdam, The Netherlands; s.liu2@uva.nl; 2Department of Medical Microbiology, Centre for Infection and Immunity Amsterdam (CINIMA), Academic Medical Centre, University of Amsterdam, 1105 AZ Amsterdam, The Netherlands; s.a.zaat@amsterdamumc.nl

**Keywords:** foodborne pathogen, persisters, bacterial spores, antimicrobial peptides

## Abstract

The occurrence of bacterial pathogens in the food chain has caused a severe impact on public health and welfare in both developing and developed countries. Moreover, the existence of antimicrobial-tolerant persisting morphotypes of these pathogens including both persister-cells as well as bacterial spores contributes to difficulty in elimination and in recurrent infection. Therefore, comprehensive understanding of the behavior of these persisting bacterial forms in their environmental niche and upon infection of humans is necessary. Since traditional antimicrobials fail to kill persisters and spores due to their (extremely) low metabolic activities, antimicrobial peptides (AMPs) have been intensively investigated as one of the most promising strategies against these persisting bacterial forms, showing high efficacy of inactivation. In addition, AMP-based foodborne pathogen detection and prevention of infection has made significant progress. This review focuses on recent research on common bacterial pathogens in the food chain, their persisting morphotypes, and on AMP-based solutions. Challenges in research and application of AMPs are described.

## 1. Introduction

The occurrence of bacterial pathogens in the food chain is an important cause of foodborne diseases and food poisoning. Both disease states cause severe public health problems and affect well-being in both developed and developing countries [1]. According to annual data from the World Health Organization, almost 1 in 10 people worldwide falls ill because of contaminated food, and nearly one-third of 42,000 food contamination-deaths occur among children under five years of age. In low-income or middle-income countries, losses accounting for up to $110 billion are linked to unsafe food. Due to the high morbidity and mortality rates, foodborne pathogens have gained great attention among researchers.

The current antibiotic crisis due to drug resistance and drug tolerance hinders the eradication of pathogens in both the food industry and in patients. Resistance mechanisms such as drug efflux and modification of drug targets are inherited traits attributable to genetic alterations [2]. In contrast, drug tolerance is a property of bacterial persister-cells, which are dormant variants of regular cells arising by non-heritable mechanisms [3]. In 1942, a small fraction within a *Staphylococcus aureus* population refractory to penicillin treatment was discovered [4], and, in 1944, the term persisters was coined for these cells [5]. Since this first discovery, scientist reported stochastic phenotypic variations conferring drug tolerance in almost every bacterial species [6].

Although a universal definition is lacking, persisters are normally described as transiently antibiotic tolerant phenotypic variants that allow a population to survive antibiotic exposure [7]. It is noteworthy that persisters are not necessarily completely and metabolically inactive. For example, intramacrophage *Salmonella* persisters secrete certain effectors to diminish host pro-inflammatory immune responses and induce macrophage polarization [8]. Persisters formed either during stress conditions (type I persisters) or are continuously generated during exponential growth (type II persisters), which allows a subpopulation to always be prepared to tolerate future stresses [9]. Clinically, persisters play a role in chronic, recurrent, and antibiotic-resilient infections. In addition, persisters contribute to antibiotic resistance development. For instance, in vitro experiments showed that formation of persisters precedes and promotes the emergence of heritable resistance [10]. Bakkeren et al. [11] reported that *Salmonella* persisters promote the spread of resistance plasmids in the gut. Therefore, a full understanding of persisters and the stimuli inducing their formation is much desired. The more we know about various persister types and their formation mechanisms, the better we can control this special phenotype to reduce its impact.

Due to the low metabolic activity of persister-cells, antibiotics fail to eradicate them. Therefore, new approaches are needed. One possible treatment for persisters is the use of natural or synthetic antimicrobial peptides (AMPs). Natural AMPs are small molecules produced by animals and plants as the first line of defense against a wide spectrum of pathogenic microbes, and by microbes to maintain their environmental niche. Currently, more than 3200 natural AMPs are recorded in The Antimicrobial Peptide Database (http://aps.unmc.edu/AP/main.php). Though the synthesis is still costly, it is possible to modify or fully synthesize AMPs to widen the scope of their antimicrobial action as well as to address natural disadvantages such as instability and potential toxicity [12]. Unlike most antibiotics, which target specific cellular metabolic processes, AMPs act via direct interaction with bacterial membranes and, if deployed to combat infections, through interaction with host immunity [13]. Additionally, several AMPs can interact with intracellular targets [14]. The synergy of AMPs with antibiotics or other antimicrobial strategies showed great potential against persisters [15]. In summary, due to their high killing efficiency and broad spectrum, AMPs are not only used as persister eradicators but in specific cases as food preservatives (such as nisin and its derivatives) and pathogen detectors.

In this review, we aim to provide an overview of recent progress and current deficiencies of controlling common foodborne pathogenic persisters and of AMP-based solutions. We first introduce major common foodborne pathogens and their huge impact on human health as well as on the food industry. Next, we describe persisting morphotypes of the mentioned five foodborne pathogens and the mechanisms of their formation. Following that, we summarize recent progress made in research on AMPs, and describe their identified modes of action and their application in persister eradication, food preservation, and pathogen detection. Finally, we provide a brief overview of the current challenges for AMPs to become anti-persister agents of the future.

## 2. Persisters of Foodborne Pathogens

Bacterial foodborne pathogens cause foodborne illness by the ingestion of pathogens themselves and/or by pathogen-produced toxins. Precisely, food-induced illness can be divided into two types: food infection and food intoxication [16]. The former is caused by swallowing foods with pathogens that grow and secrete a toxin once inside the human body, whereas the latter is due to eating foods containing toxins produced by microbes. In this review, we focus on persisters of common foodborne pathogens, which are also intensively used for persister research. These include the Gram-negative pathogens *Salmonella* and *Escherichia coli*, and the Gram-positive species *Staphylococcus aureus*, *Listeria monocytogenes*, and *Bacillus cereus.* These pathogens and their toxins can come from diverse sources including polluted water, soil and air, contaminated raw materials, and non-hygienic food processing and storage (Figure 1). Environmental stress conditions may lead to the generation of persisting morphotypes of these pathogens in the food chain. Note that recent data on the globally leading cause of human bacterial gastroenteritis, *Campylobacter jejuni*, show that this bacterium is able to form persister-cells upon exposure to the antibiotics’ ciprofloxacin and penicillin G [17,18,19].

### 2.1. Common Bacterial Foodborne Pathogens

*Salmonella* are Gram-negative, rod-shaped, and facultatively anaerobic bacteria that commonly cause food contamination. According to the European Food Safety Authority, in 2018, almost one in three foodborne outbreaks in Europe was due to *Salmonella* found in raw or undercooked eggs and meats, contaminated fresh produce, or in dairy products. There are over 2500 identified *Salmonella* serotypes grouped into two species: *S. bongori* and *S. enterica.* Clinically, almost all *Salmonella* populations that cause human and domestic diseases belong to *S. enterica* subspecies *enterica*. Among that, the most common pathogens are *S. enterica* serotypes Typhimurium and Enteritidis [20]. The majority of *S. enterica* serovars are nontyphoidal *Salmonella*, which cause milder nontyphoidal salmonellosis with symptoms like diarrhea, fever, vomiting, and abdominal pain [1]. Typhoid fever, a global problem affecting millions of people and causing over 200,000 deaths annually, is caused by the intracellular and human host-restricted pathogen *S. enterica* serovar Typhi (*S.* Typhi) through the action of a *Salmonella* cytolethal distending toxin among others [21].

*Escherichia coli* is a widespread gram-negative, rod-shaped bacterium colonizing the guts of humans and animals. Although most *E. coli* strains are harmless, several pathogenic species can cause severe illness. These species cause infection primarily through the fecal–oral route, and include (1) enteropathogenic *E. coli* (EPEC), which is the main cause of infantile diarrhea outbreaks and recurrent diarrheal symptoms [22], (2) enterohemorrhagic *E. coli* (EHEC), often times cause hemorrhagic colitis and hemolytic uremic syndrome [23], (3) Shiga toxin-producing *E. coli* (STEC), a human pathogen leading to several diseases such as bloody diarrhea and even death (around 25% of cases) [24], (4) enteroinvasive *E. coli* (EIEC), that are highly invasive to colon epithelial cells causing bacillary dysentery [25], (5) enteroaggregative *E. coli* (EAEC), which is one of the most prevalent causes of acute and persistent diarrhea due to contaminated foods [26], and (6) enterotoxigenic *E. coli* (ETEC) that infect travelers and children in developing and semitropical areas and cause diarrhea-associated symptoms [27]. Foodborne *E. coli* outbreaks are commonly associated with fresh produce, meat products, and unclean water caused by a contaminated environment and/or non-hygienic food processing.

*Staphylococcus aureus* is a Gram-positive, coccoid bacterium. Foodborne *S. aureus* strains are food intoxication bacteria producing thermal resist toxin, which is responsible for the most prevalent foodborne intoxication worldwide named Staphylococcal Food Poisoning (SFP) [28], leading to nausea, violent vomiting, abdominal cramping, fever, and diarrhea. *S. aureus* colonizes mostly the human nose but also skin, hair, and mucous membranes. Even though most of the ubiquitous *S. aureus* populations on the human body are indicated to be harmless [29], they are responsible for high frequency of SFP due to improper food handling. Except for unhygienic food processing, contaminated raw milk and soft cheese that originate from animals suffering *S. aureus*-induced mastitis are also possible inducers of SFP [30]. Note that, upon gaining access to the body through an injury on the skin of mammals, certain *S. aureus* strains can also cause skin and joint infections, pneumonia, sepsis, endocarditis, and medical device-associated infection [31,32]. These events are not the central topic of this review.

*Listeria monocytogenes* is a Gram-positive, rod-shaped, facultative intracellular foodborne pathogen and the inducer of a relatively rare disease known as listeriosis. This food-borne illness is characterized by a high hospitalization and fatality rate (up to 30%) [33]. This potentially lethal infection is associated with symptoms ranging from mild gastroenteritis to fatality cases and the bacteria have higher infectivity and virulence in vulnerable populations like pregnant women, children within five years of birth, the elderly, and patients with deficient immunity [34]. *L. monocytogenes* can replicate at a wide range of temperatures (2–45 °C) and divergent environments including mammalian cells, and is able to survive under ubiquitous stresses such as high acidity and high salinity [35]. This makes them extremely resistant to inactivation treatments both in vitro and in vivo. *L. monocytogenes* is commonly found in processed and cold-stored food such as meat products and dairy products. Recently, fresh produce has also been reported to be a contributor to listeriosis [33].

*Bacillus cereus* is a Gram-positive, aerobic facultative, spore-forming bacterium that has been ubiquitously found in soil, water, vegetables, meat, and milk [36]. While certain *B. cereus* strains are used as a growth promoter or biocontrol against microbial illness, or as probiotics for mammals, other *B. cereus* strains lead to food poisoning [37]. Pathogenic *B. cereus* mainly causes two kinds of gastrointestinal diseases: (1) emesis linked to cereulide, a heat-stable and acidic-stable non-ribosomal depsipeptide (pre-) formed in the food matrix, commonly associated with starchy foods like pasta and rice [38], and (2) diarrhea induced by enterotoxins produced in the small intestine, namely hemolysin BL (HBL), non-hemolytic enterotoxin (NHE), and single proteins such as cytotoxin K and enterotoxin T [39]. Although diseases caused by *B. cereus* are generally mild and self-limiting, severe and lethal cases have been reported [40,41]. As the third most frequent foodborne outbreak-related pathogen in Europe [42], *B. cereus* is receiving increasing attention.

Finally, as indicated, we now know that *C. jejuni* strains can also form persister-cells upon ciprofloxacin treatment while such an observation was not made upon exposure to the beta-lactam antibiotic ampicillin [19]. However, the mechanism of *C. jejuni* persister formation is largely unknown yet. However, recent experiments suggest that *C. jejuni* cells could undergo a remodeling of the electron transport chain in order to moderate membrane hyperpolarization and intracellular alkalization. These events might jointly reduce the antibiotic efficacy and potentially assist in persister-cell formation [18].

Foodborne pathogens and their persisting forms cause severe food safety and quality challenges to food manufacturing in general and the food industry in particular. The existence of persister cellular forms, which can evade antimicrobial action and regrow after stress removal, makes it difficult to eliminate those pathogens. Although the first discovery of persister-cells dates back 78 years, the contribution of these cells to infection has been underestimated for a long time, until they were clinically isolated from patients who experienced fatal relapse of infection after antibiotic failure [43,44]. Convincing and convergent experimental evidence has indicated that persisters hold responsibility for therapy failure and recalcitrance of diseases despite prolonged antibiotic treatment [6,45,46].

### 2.2. Definition of Persisters

In 1944, Bigger termed persisters a subset of cocci of a *Staphylococci* population that survived inactivation with penicillin as a non-growing dormant cellular phenotype without being genetically resistant [5]. Since then, persisters are mainly considered as physiologically dormant cells that can evade antibiotic treatments. However, recent research has questioned this concept. First of all, some types of persisters are correlated with active metabolism. For instance, increased metabolic activity leads to a higher level of persisters in a stationary phase population [47]. Intracellular *Salmonella* persisters are able to secrete compounds to disturb the host immune system and avoid being killed [8]. Lastly, not all non-growing cells are persisters [48].

In 2019, Balaban, together with other 120 persistence-involved researchers, provided a common definition of antibiotic persistence and persisters [49]. In general, antibiotic persistence (1) is a phenotypic feature of a subset of a bacterial population that can survive bactericidal antibiotic exposure, (2) is experimentally characterized by biphasic kill curves, which means that, while a large part of the population is killed by an antibiotic, a smaller fraction survives for a longer time, and (3) is a non-heritable and a population-level phenomenon. Meanwhile, persisters are defined as tolerant cells isolated from persistent populations.

According to the distinct way of persister formation, persisters are generally divided into two types [9]: Triggered (type I) persisters induced by a myriad of stress conditions, such as exposure to antimicrobial agents, energy limitation, acid, immune cells, and compounds, which are spontaneous persisters (type II) that are continuously generated during growth. Yet, instead of exhibiting a unified antibiotic resistance profile in a single given condition, persisters show heterogenous abilities to withstand antimicrobials due to different ways of obtaining persister-cells [50,51]. This may indicate that a quiescent target or dormancy is not the only strategy used for surviving. Different mechanisms of persister formation lead to divergent persisters. The definition of a persister needs to be extended to other fatal stress conditions. To sum up, the current definition of persisters is still evolving.

In this review, we define persisters as a genetically susceptible and growth-arrested subpopulation that have a non-inheritable capability of surviving under possibly fatal conditions and can regrow into normal sensitive bacteria after stress removal. In this case, divergent phenotypes fitting this definition can be termed persisters (Figure 2), such as in planktonic bacteria stress-induced growth-arrested persisters as well as stochastically-formed persisters, biofilm-contained persisters, small colony variants (SCVs), unstable L-form cells, intracellular persisters, and bacterial spores.

### 2.3. Persister Types and Mechanisms of Formation

Slow growth/no-growth is a clear characteristic of persisters, and conditions that lead to bacterial growth arrest strongly promote persister formation. Growth arrest occurs in energy-limited conditions. Conlon et al. [52] discovered that stationary phase *S. aureus* cells, which displayed an intracellular ATP drop, showed a 325-fold increase in persister formation, and a 100–1000 times higher level of antibiotic tolerance compared to their exponential phase counterparts. Similarly, Shan et al. [53] demonstrated that a low level of ATP is the underlying explanation of *E. coli* persister production. However, Wang et al. [54] reported that *S. aureus* persisters formed in a stationary phase are associated with low membrane potential and ATP depletion is merely an indirect outcome. In addition, nutritional deficiency inside macrophages is regarded as a cause of intramacrophage *Salmonella* persister formation [55]. Long-term starvation results in sporulation in spore-forming *B. cereus* [56]. Except for energy limitation, artificial/stress-induced growth arrest also leads to a higher persister level. Deficiency of heme leads to higher *S. aureus* persister levels under energy stress and antibiotic treatment, which may be due to the important role of heme in the respiration chain and energy production [57]. Moreover, signal pathways depending on guanosine pentaphosphate/tetraphosphate (ppGpp) and Toxin-antitoxin systems (TA systems) lead to growth arrest due to various metabolism-halting mechanisms (summarized in Section 2.3.1 and Section 2.3.2), making them the most important molecular mechanisms of persister formation. Pontes et al. [58] argued that slow growth, irrespective of the way this is induced, is the ultimate indicator of persister formation, at least in an *S.* Typhimurium population. In general, even though the underlying explanation of growth arrest-induced persisters is still confusing, growth arrest is not only one of the most distinctive features, but also a controversial inducer of persisters. It should be noted that the reduction of growth and metabolic activity is not a sufficient trigger for persister formation as not all “slow-growing cells” show persistence [48]. In the next paragraphs, we will discuss the details of the different ways to develop a persister state.

#### 2.3.1. Guanosine Pentaphosphate or Tetraphosphate (ppGpp) and Persisters

ppGpp is an important stress alarmone in bacteria, associated with a stringent response, energy metabolism, virulence, tolerance, and persistence [59]. Under specific stress conditions such as amino acid limitation and antibiotic treatment, ppGpp can be rapidly produced by the RelA/SpoT homologue family and strongly inhibit macromolecular synthesis [60]. Several lines of research provide firm evidence that ppGpp makes a major contribution to persister formation (Figure 3). Maisonneuve et al. [61] claimed that stochastic expression of ppGpp is the key cause of type II persister formation in exponentially growing *E. coli* populations. Song et al. [62] proposed an ppGpp ribosome dimerization persister (PRDP) model that ppGpp induces *E. coli* persister formation by directly halting ribosome action through the ribosome modulation factor, a hibernation promoting factor, and a ribosome-associated inhibitor. Furthermore, ppGpp also participates in the toxin-antitoxin (TA) system-induced *E. coli* persisters. Svenningsen et al. [63] showed that ppGpp accumulation beyond a threshold induced by tRNA limitation results in higher persister levels in an *E. coli* population. However, there is no clear correlation between ppGpp expression and persistence at a single cell level.

In addition to *E. coli*, ppGpp also participates in persister formation in other species. Helaine et al. [55] discovered that the formation of intramacrophage *Salmonella* persisters is associated with the ppGpp/lon protease-based TA system and is specifically triggered by acid and an energy-limited host cell environment. Recently, Fung et al. [64] proposed that ppGpp-induced GTP depletion is a shared metabolic pathway of both triggered and spontaneous persisters (non-spore form) in a strain of the Gram-positive model species *B. subtilis*. In this process, different pathways of persister formation are activated by various ppGpp synthetases. Clinically, Gao et al. [65] found that *S. aureus* SCVs isolated from patients who suffer from persistent and recurrent infection have more ppGpp accumulation.

#### 2.3.2. Toxin-Antitoxin System Induced Persisters

TA systems are defined as ubiquitous small operons containing two genes that separately express a stable toxin molecule to slow down/block certain metabolic processes and an unstable antitoxin to neutralize this toxicity. Under certain stress, the antitoxin will be selectively degraded, which consequently leads to toxin accumulation and growth arrest that ultimately contributes to persister formation [66]. Currently, there are six kinds of TA systems (type I–VI) and persistence-associated TA systems belong to the first two types (Figure 4) [66,67]: (1) a type I TA system contains a small RNA (sRNA) antitoxin inhibiting the translation of toxin that decreases ATP synthesis by disrupting membrane potential, (2) the best-studied type II TA system includes a DNA-binding protein antitoxin that can repress toxin gene transcription and/or directly inactivate protein toxin by forming a TA complex. This type II toxin inhibits several essential metabolic processes such as DNA or protein synthesis. Most type II toxins can also co-repress transcription with an antitoxin [68].

*E. coli* is an important species to study the function of type II TA systems in persister formation. The first discovered gene that directly associated with persister levels is *E. coli hipA* gene encoding high persister protein A. *HipA* is part of a type II TA locus, *hipBA*, with its inhibitor gene *hip B* [69]. The HipA toxin attenuates cell growth by inhibiting glutamyl tRNA synthetase (GltX), and then stimulates ppGpp synthesis and, ultimately, promotes persister formation [70,71]. In addition, *hipA*-induced *E. coli* persisters highly rely on ppGpp [72]. In addition to *hip BA*, there are 10 other type II ppGpp-independent TA loci in *E*. *coli* that encode mRNases involved in the reduction of cellular metabolic activity [73]. In 2013, Maisonneuve et al. [74] provided strong evidence that *E. coli* mutants without these 10 TA loci display sharply decreased persister levels. However, it was proven in 2017 that these mutant cells were contaminated by a φ80 bacteriophage and the article was retracted [75]. Later, Goormaghtigh et al. [76] reassessed the role of mRNases-encoding TA loci and demonstrated that they are not directly linked to antibiotic persistence. Despite this, evidence still suggests an essential role of TA systems in *E. coli* persister formation. For example, Tripathi et al. [77] proved that toxin MazF-mediated growth arrest leads to high levels of persisters under antibiotic exposure. Harrison et al. [78] revealed the importance of the *yafQ* toxin gene in inducing persisters in antibiotic-exposed biofilms.

Although reports about *E. coli* persisters and their TA systems are conflicting, positive relationships between the TA toxin and persister formation are proposed in *Salmonella*. Helaine and colleagues discovered that the majority of 14 ppGpp-dependent type II TA operons play key roles in intramacrophage persister formation [79]. Later, acetyltransferase toxin TacT and three TacT-like toxins were revealed and their ability to induce persisters by inhibiting translation was further proven [80,81]. On the opposite side, TA II systems do not have a clear effect in *S. aureus* [52] and *L. monocytogenes* [82]. In addition to type II TA systems, type I TA systems have also been proven to mediate persister formation. The overexpression of type I TA toxin *tisB* activated by the SOS response lead to a significant increase in *E. coli* persister formation [83]. In 2020, Habib et al. [84] first reported a novel type I TA system (termed TMCS) in *S. aureus* that promotes persister formation by inhibiting the efflux pump NorA under antibiotic treatment. In summary, TA systems are likely linked to the molecular physiology of persister formation.

#### 2.3.3. Persisters in a Biofilm

Biofilms are major contributors to cross-contamination in food processing because they are tolerant to several stresses and extremely difficult to eliminate [85]. Within a biofilm, cells protected by self-produced matrix material are less sensitive to antibiotics, high salinity, sanitizer, and a host immune system compared to planktonic cells [86]. Biofilms can stick on surfaces of stainless steel of storage and pipe systems. In such a physiological state, these bacteria can pose persistent food safety issues in the food industries [87]. Biofilm-related infections have led to increased morbidity and mortality, and cost nearly $450 million per year [88]. As a small part of the biofilm population, persister-cells (including bacterial spores) are believed to be the essential explanation of biofilm antibiotic tolerance [89]. This is also of importance should a host become infected with (foodborne) pathogenic microorganisms. While the majority of sensitive cells in a biofilm are killed by antibiotics, persister-cells can survive. Once the lethal condition for the bacteria is relieved, persisters within a biofilm can revert into toxin-secreting vegetative cells that can repopulate industrially relevant biofilms and/or medically relevant biofilms causing human infection. For instance, biofilms formed on gallstones act as reservoirs of *S. typhi* persisters, contributing to the spread of typhoid fever [90].

Within the biofilm microenvironment, an activated SOS stress response and accumulated ppGpp are important inducers of persister formation [91,92]. This makes biofilms of *E. coli, Salmonella*, *Bacillus*, and *Staphylococcus* populations contain more persisters than liquid cultures, and these persisters can maintain their phenotype for up to four weeks on antibiotic-containing nutrient-rich medium [93]. For *E. coli*, the TA system MqsR/MqsA directly regulates biofilm formation and the persistence level through toxin MqsR and CspD as well as through the quorum sensing signal autoinducer-2 (AI-2) [94]. In *S. aureus* and *Salmonella* populations, the master virulence regulator Agr QS has been proven to serve as a repressor of persister formation by regulating a family of virulence factors named phenol soluble modulins (PSMs) [95]. The latter play an important role in biofilm formation [96]. In general, persister formation and biofilm formation reinforce each other with respect to (environmental) stress resistance.

#### 2.3.4. Small Colony Variants

SCVs are defined as slow-growing pin-prick-sized colonies that are less sensitive to antibiotic treatments compared to wildtype pathogens and can revert to a normal phonotype after stress removal [97]. Since the first report about associations between SCVs and persistent infection in 1995 [98], researchers have found that SCVs exist in a wide range of bacteria and are linked to several chronic diseases [99]. In addition, SCVs can promote pathogenicity by promoting biofilm formation [81], presumably facilitated by overexpressing adhesins [100] and by impairing host immunity via necroptosis induction [101].

It was found that environmental stress, antibiotic therapy, and adverse physiological conditions inside eukaryotic cells, are inducers of SCVs. For example, long-term cold stress raises the proportion of *S. aureus* SCVs with thicker and diffuse cell walls [102]. Exposure to quinolones [103] and aminoglycosides [104] can also lead to *S. aureus* SCVs. Vulin et al. [105] observed that *S. aureus* SCVs isolated from clinical infections and culture under in vitro stress conditions, displayed a prolonged lag time. Further research showed that such a prolonged lag time that can be induced after is antibiotic exposure in vitro. This suggests that antibiotics may increase SCV formation, and, thus, the risk of persister formation in the clinic.

Though most SCVs-related studies use *S. aureus* as model strains, SCV formation is also used as a survival strategy by *Salmonella*. SCVs form in *Salmonella* planktonic and biofilm cultures when exposed to a high concentration of ciprofloxacin and ceftazidime [106]. Moreover, the formation of SCVs is recognized as part of persistent *Salmonella* [99] infection. Such SCVs, localized within host cells, can last for several weeks and contribute to a relapse of infection.

#### 2.3.5. L-Form Bacteria

Another type of persister is the unstable L-form cell. L-forms were first observed and named by Emmy Klieneberger in 1935 [107]. Under certain stress conditions, bacteria can form stable and unstable L-form populations presenting either as protoplasts with no cell-wall or spheroplasts with an incomplete cell-wall. While stable L-form populations are generated by genetic mutations, unstable ones are phenotypic variants that can revert to normal walled cells after the removal of the inducer. Unstable L-form *S. aureus* can be produced in culture media including special contents, such as osmo-protective compounds as well as serum and cell wall biogenesis inhibiting antibiotics [108]. This presents unique features such as pleomorphism, slow growth, and the formation of “fried egg” colonies on soft agar [109]. They are associated with recurrent and chronic diseases that are resistant to cell-wall targeting antibiotics methicillin and oxacillin [110]. Han et al. [108] provided evidence that the *glpF* gene and *glpK* gene involved in glycerol uptake and cell membrane synthesis are essential for unstable L-forms formation and persister level in an *S. aureus* population. A multitude of studies reported stable L-form cells in *Bacillus subtilis* and *E. coli* populations [111], but little is known about unstable L-form cells, partly due to the difficulty to separate and maintain these cells.

#### 2.3.6. Intracellular Persisters

*S*. Typhimurium is able to invade macrophage cells to escape from antibiotic treatment and the host immune system. During this process, vacuolar acidification and nutritional limitation circumstances stimulate a portion of cells to switch to a metabolically-active persister phenotype through a TA system [79]. Another typical intracellular pathogen *L. monocytogenes* has long been considered as a cytosolic bacterium [112]. However, it has been recently revealed that subpopulations of *L. monocytogenes* can switch from a cytosolic actin-based motile lifestyle into a non/slow-growing persister state in vacuoles, which may contribute to the asymptomatic period of listeriosis [113] and chronic infections. During this process, spacious Listeria-containing vacuoles (SLAPs) in phagocytic cells of severe combined immunodeficiency mice [114] and lysosome-like vacuoles termed Listeria-Containing Vacuoles (LisCVs) in in vitro cultured hepatocytes and trophoblast cells are seen [115]. The environmental conditions in these vacuoles are considered to promote persister formation. However, in vivo evidence, such as LisCVs formation and clinical isolation of *L. monocytogenes* persisters, are still scarce but available [113,116].

#### 2.3.7. Sporulation

Under extreme conditions—like long-term starvation, heat stress, the presence of toxic compounds, or radiation, *Bacillus* is able to form spores that can survive for decades through a process known as sporulation (Figure 5). Compared to vegetative cells, spores are more tolerant to harsh environments, such as acid (pH value: 1–5.2), an elevated temperature (2 min at 95 °C or 32 min at 85 °C), antibiotic exposure, cold, irradiation, food preservatives, antibiotic treatment, and stomach passage [117], making them very hard to eradicate. Noticeably, spores from pathogens such *B. cereus* can be easily transferred from soil or water to food and remain in food products for a long period [118]. When conditions become favorable, spores can germinate and grow out into vegetative cells. Germination can occur at temperatures ranging from 5–50 °C in cooked food and −1–59 °C in culture media, which further increases the infection risk of *B. cereus* [119]. It was found that floating *B. cereus* cells maintain a stationary phase with high infectivity to host cells without forming spores [37]. In contrast, spores within biofilms on the food contact surface account for a potentially large part of *B. cereus* infection after cleaning and disinfection procedures in the food chain. These two co-existing forms of the bacterium allow *B. cereus* to efficiently survive in different environmental niches.

## 3. Antimicrobial Peptides (AMPs)

AMPs are natural or artificial short peptides (5–40 amino acids) containing multiple hydrophobic residues. Natural AMPs are ubiquitously synthesized by microbes, plants, animals, and humans as a first-line host defense with potent anti-bacterial, anti-virus, and anti-fungal activities. Since Gramicidin S was initially extract from *Bacillus brevis* in 1944 [120], 3236 AMPs from six kingdoms have been listed in ‘The Antimicrobial Peptide Database’ up to now. Structurally, most AMPs can be categorized into four major groups, namely α-helical, β-sheet, loop, and extended AMPs, in which the first two are most common in nature and α-helical AMPs are the most researched ones to date [121]. There are other AMPs that either have two or more different structural components, or change structures during their interaction with their targets, such as indolicidin [12]. While AMPs have remarkably different sequences and structures, most AMPs act against bacteria via a membrane disturbance. Compared to antibiotics that destroy pathogens by halting specific metabolic processes, these membrane-targeting AMPs do not rely on bacterial metabolic activity to exert their antimicrobial effect. Therefore, they are able to effectively inactivate vegetative pathogens as well as their persister-cells with a low risk of inducing resistance. In addition, certain AMPs can also inactivate pathogens by inhibiting intracellular metabolism or modulating the host immune system. Here we discuss of a number of AMPs (Appendix A) the mode of action and applications

### 3.1. Mechanisms of Action

#### 3.1.1. Membrane-Targeting AMPs

A commonly known antibacterial mechanism of AMPs is the non-receptor mediated membrane-lytic activity that relies on two features: net positive charge to ensure the interaction with negatively charged compounds on the bacterial envelop (e.g., lipopolysaccharides (LPS) of Gram-negative bacteria, teichoic acids of Gram-positive bacteria, and phospholipids of the bacterial membrane), and amphipathic propensity, which allow AMPs to fold into both hydrophobic and hydrophilic structures [122]. After attaching to the bacterial cytoplasmic membrane, a wide range of AMPs insert into the bilayers and subsequently cause membrane structural disruption and permeabilization. In vitro studies with artificial membranes have led to three proposed models [123,124,125], namely (1) the toroidal-pore (Figure 6A) that is a continuously toroidal channel formed through AMPs-induced bending of the bilayers and proposed for PGLa and magainins, (2) the carpet model (Figure 6B) where AMPs induce membrane disintegration by various transient holes like a carpet, used by AMPs like caerin 1.1 and cecropins, (3) the barrel-stave pore (Figure 6C), which is a highly ordered cylindrical water pore surrounded by peptides and is linked to hydrophobic peptides such as alamethicin.

Recently, it was found that SAAP-148, an artificial AMP derived from LL-37, strongly disturbs the hydrophobic core, which can lead to membrane thinning and rapid permeabilization [126] (Figure 6D). It was proven that amphipathic cationic AMPs (thrombocidin-1 derived peptides TC19 and TC 84, and the designed AMP BP2) can induce extensive fluidic domains in the *B. subtilis* membrane [127] (Figure 6E), and, in addition, can cause an irregular or shrunken inner membrane of geminated spores [128] (Figure 6F). Other mechanisms have also been proposed mainly for small, unstructured, or compact peptides that cannot directly span the membrane to form pores, such as molecular electroporation, sinking raft, interfacial activity, and lipocentric pore formation models [129]. AMPs-induced membrane perturbation results in membrane depolarization, cellular contents leakage, and intracellular disruption, which further leads to cell death [123].

#### 3.1.2. AMPs Affecting Intracellular Physiology

In addition to membrane targeting AMPs, there is growing evidence that certain AMPs directly interact with essential intracellular mechanisms, such as principle macromolecular biogenesis, cell wall biosynthesis, and cell division [14] after they have entered the cytoplasm through membrane disturbance, forming transient pores or receptor-mediated processes [130]. For instance, buforin II uses a proline hinge as a cell-penetrating promoter and then inhibits DNA and RNA synthesis in *E. coli* [131]. The 13-amino acid indolicidin translocates by membrane permeation, then stabilizes duplex DNA, and, consequently, halts DNA replication and transcription. Its synthetic 13-residue variant CP10A interacts with the membrane and shows inhibitory ability against DNA, RNA, and protein synthesis in treated *S. aureus* [132]. Microcin J25 isolated from *E. coli* fecal strain AY25 can kill *Salmonella* by binding and obstructing the second channel of RNA polymerase [133] and inhibiting the cell division process of other *E. coli* cells [134]. eNAP-2 generated by equine neutrophils can eradicate *E. coli* by non-covalently binding to the bacterial serine proteases subtilisin A and proteinase K [135]. Nisin, which is the most investigated lantibiotic, is commonly used against Gram-positive pathogens like *S. aureus* and *Bacillus* spores, and targets lipid II, inhibiting cell wall synthesis [136].

#### 3.1.3. Immunomodulatory AMPs

Except for inherent antimicrobial activity, certain AMPs (also termed as host-defense peptides (HDPs)) are found to modulate the host immune system in vivo and in vitro by manipulating immune and pro-inflammatory/anti-inflammatory responses, such as chemo-attraction and programmed cell death, contributing to the clearance of pathogens, as well as their persisters. Moreover, evidence shows that the alterations of HDPs levels are related to diverse (multiple) infection events and inflammation. This phenomenon can potentially be used as a biomarker for such diseases [137]. In terms of foodborne pathogens causing diarrhea and infectious colitis (such as *E. coli*, *Salmonella* and *Listeria*), DHPs produced by colonic epithelial cells and leukocytes are crucial parts of the innate immune response in the colon against these enteropathogenic bacteria [138].

In humans, cathelicidin and β-structured defensins are two primary families of HDPs, related to both innate and adaptive immunity [139]. There is only one human cathelicidin—hCAP18, secreted by a range of cells including mucosal epithelial cells, mast cells, and various immune cells [140]. LL-37 is cleaved from the C-terminal of hCAP18 by serine proteases [137], and is one of the best studies human AMPs. As an immunomodulatory AMP, LL-37 cannot only rapidly kill a wide range of pathogens by creating transmembrane pores, but also work as an “alarmin” that modulates immune mechanisms [141]. Specifically, LL-37 interacts with more than 16 protein receptors and, consequently, putatively modulates expression of hundreds of genes and thousands of second receptor proteins. As a result, it modulates various pro-inflammatory and anti-inflammatory activities depending on the cell type and inflammatory stimuli [139].

Similar to cathelicidin, defensins possess both effective bactericidal function and relatively modest regulatory activity in inflammation and immunity. Human defensins are cysteine-rich peptides, including two subgroups: α- and β-defensins. α-Defensins, also known as human neutrophil peptides (HNPs), are produced by azurophilic neutrophil granules (HNP1-4) or by Paneth cells (HD5-6) in intestinal mucosa. In comparison, β-defensins (hBDs) are longer peptides with more lysines, mainly released by epithelial cells and immune cells such as macrophages. For now, at least 48 human hBDs gene are found and six hBDs have been characterized (hBD1-6) [142], regulating the immune system by inducing chemotaxis secretion and promoting the maturation and activation of leukocytes [143].

### 3.2. Application of AMPs Against Foodborne Pathogens and Persisters

#### 3.2.1. AMPs Used in Food

Bacteriocins are small, heat-stable, ribosomally-synthesized AMPs or proteins generated by many bacteria, prominently including lactic acid bacteria (LAB). Due to their characteristics of effectively eradicating foodborne pathogens and persisters as well as being harmless and easily digested in the human digestive tract, bacteriocins and corresponding LAB are traditionally used as food preservers [144]. Unlike other AMPs, most bacteriocins have a relatively narrow spectrum of antimicrobial activity against closely-related bacteria [145]. Normally, there are three categories of bacteriocins-based food preservation strategies: antimicrobial solution with partially purified bacteriocins, bacteriocin-containing fermented products, and bacteriocin-producing microbes [146]. Structurally, bacteriocins can be divided into four groups in which class I–II are peptides smaller than 5 kDa and 10 kDa, respectively, and class III–IV are antimicrobial proteins that are not mentioned in this review.

Class I bacteriocins, also known as lantibiotics, are composed of post-translationally modified peptides with unusual amino acids like lanthionine [147]. With broad-spectrum and efficient killing activity, nisin is the most famous class I bacteriocin used to improve food safety. Nisin is produced by *Lactococcus lactis*, and acts against Gram-positive pathogens including *L. monocytogenes* and *S. aureus* by binding to lipid II and consequently halting cell wall biosynthesis as well as making holes in the membrane [136]. In addition, Omardien et al. [128] discovered that nisin inhibits *Bacillus* spore outgrowth via its perturbing effect on the inner spore membrane. Another class I bacteriocin, lacticin 3147 is used by a veterinarian cooperatively with nisin against mastitis in cows. Besides, lacticin 3147 shows higher *B. cereus* killing efficiency than nisin [148]. Class II bacteriocins (non-lantibiotics) contain non-modified peptides and are divided into five subgroups IIa–IIe. Among them, Class IIa pediocin-like bacteriocins from several Gram-positive bacteria are majorly used agents to prevent *L. monocytogenes* by targeting bacterial mannose phosphotransferase [145]. Nisin and pediocin have been approved for commercial use in food, which further led to steadily growing usage of bacteriocins and bacteriocin-producing LAB in the dairy industry to prevent foodborne pathogens [149].

#### 3.2.2. AMP-Based Detection of Foodborne Pathogens and Persisters

Compared with traditional detective strategies for foodborne pathogens including colony counting, the polymerase chain reaction (PCR) test, immunology-based assays, and test kits, biosensors possess significant advantages such as high sensitivity, efficiency, and convenience [150]. AMPs can also be used to develop biosensors due to their stability, flexibility, low-cost, easy synthesis, and high affinity [151]. Most importantly, compared to commonly used aptamers-based and antibiotics-based biosensors, AMPs-based biosensors can effectively attach on the outer membrane of pathogenic bacteria including persisters, and provide real-time food pathogens detection. Compared to antibody-based sensors, AMP-based biosensors are more cost-effective. It was speculated that the attachment of the immobilized AMPs to the bacteria is achieved via electrical and hydrophobic interaction between AMPs and the bacterial membrane [152], but the interactive mechanism has only been sparsely studied in the biosensing field.

For AMP-based biosensors, availability and sensitivity of selected AMPs are of great importance. Mannoor et al. [153] designed a biosensor with magainin I immobilized on microelectrodes to detect Gram-negative pathogens and achieve a limit of detection (LOD) of 10^3^ cfu/mL for *E. coli*. Later, Jiang et al. [154] optimized Mannoor’s module by changing magainin I into Colicin V, thus, reducing the LOD of *E. coli* to 10^2^ cfu/mL. For Gram-positive pathogens, Etayash et al. [155] successfully proposed a biosensor containing immobilized Class IIa bacteriocins leucocin A with a LOD for *L. monocytogenes* of 10^3^ cfu/mL. A colorimetric biosensor based on HRP-AMPs (AMPs conjugated with horseradish peroxidase as a detection probe) can decrease the LOD for *E. coli* O157:H7 to 13 cfu/mL [156].

Moreover, the integration with emerging technologies such as nanomaterials and microfluidics can further improve the efficiency and portability of AMPs-based biosensors [152,157]. Therefore, AMPs are potential alternatives to develop productive and sensitive biosensors for detecting foodborne pathogens as well as persisters.

#### 3.2.3. Eradication of Persisters in Infection

Even though persister-cells can survive in stress conditions such as low nutrition and antibiotic exposure, they still need an intact membrane to remain viable. In this case, the ability to perturb the bacterial membrane makes AMPs powerful drug candidates against persisters. In addition, extensive research proved that many membrane-targeting AMPs show anti-biofilm activity, which also contributes to persister eradication. For instance, LL-37 displays broad-spectrum bactericidal activity by membrane-perturbing, pore-forming, and biofilm removal activities [158]. The artificial AMP SAAP-148, derived from LL-37, has high efficiency against *S. aureus* persisters as well as its biofilms without resistance selection [126]. Chen et al. [159] reported that cationic AMPs containing a varying number of arginine and tryptophan repeats effectively kill *E. coli* persisters and significantly enhance the susceptibility of biofilms to antibiotics. Human blood platelet-based TC-19, thrombocidin-based TC-84, and the designed peptide BP2 can inhibit the outgrowth of germinated *Bacillus* spores by perturbing their inner membrane [128]. Except for directly killing persister-cells by targeting cellular membranes or biofilms, novel AMPs were developed in silico to inhibit TA complexes that induce persister formation, and are proposed as a potential method against *L. monocytogenes* persisters [160].

The combination of AMPs and other bactericidal methods is another, potentially synergistic, way to combat persisters. Chen et al. [159] discovered that, while using AMP (RW)4-NH2 alone can kill up to 99% *E. coli* persisters in planktonic culture and 98% persistent cells in a mature biofilm, the synergy of ofloxacin and (RW)4-NH2 can completely eradicate viable cells including persisters. Based on the cell-penetrating characteristic of intracellular AMPs, Schmidt et al. [161] re-engineered antibiotics by adding 12 residues AMP on tobramycin to give it a membrane-penetrating activity. This composite antimicrobial agent named pentobra shows high killing efficiency against *E. coli* and *S. aureus* persisters. Rishi et al. [162] illustrated that the synergy of nisin and ampicillin is a successful method to kill *S*. Typhimurium persisters in the presence of mannitol, which is used to restore the antimicrobial susceptibility of persisters. Hence, the combination of AMPs, antibiotics, and metabolizable sugars also provides a potential strategy for persister eradication. Furthermore, the conjunction of nisin and physical methods, such as high pressure or high temperature, to control microbial contamination have a demonstrated stronger ability to eliminate recalcitrant *Bacillus* spores [117].

### 3.3. Challenges with AMPs Application

A growing number of AMPs have been proposed as effective agents that could be potentially used as not only bactericidal products, anti-persister agents, food additives, and foodborne pathogen detectors, but also immunity regulators and wound healers. Actually, a range of AMPs are proven as anti-infective agents and are currently assessed in the last clinical trial stage trail [163]. However, due to the complexity of application of AMPs, only around 60 AMPs are approved as therapeutic peptides [164] and 7 AMPs are permitted by the U.S. Food and Drug Administration for now [165]. Inherent disadvantages that hinder the application of AMPs in both food industry and clinical practice against foodborne illness are the instability in certain conditions like complex media with proteases and acid presence, short half-life, potential toxicity, and low metabolic stability [12]. For example, AMPs drugs are prone to be degraded by proteolytic enzymes in oral, tissue, or blood plasma [166], which may explain the phenomenon where some AMPs that are active in vitro require a higher dose to effectively eradicate pathogens and persisters in infectious animal models. The other possible explanation is that persisters induced by different triggers may show divergent properties and abilities against AMPs and other antimicrobial agents, which need to be further researched.

Moreover, although AMPs have a much lower propensity of inducing resistant cells compared to antibiotics, AMP resistance has been observed in certain bacteria to be mediated by reduced interaction, membrane modification, increased efflux, proteolytic degradation, and mutation [122]. For example, a variant Lipopolysaccharides (LPS) modification can be found in *S.* typhimurium populations under stress conditions, which leads to lower interaction with AMPs [167]. Last but not least, AMPs generally show broad-spectrum antimicrobial activities and have relatively low specificity to target on one certain species, which may affect the application of AMPs for systemic treatment [168] and the specificity of AMPs-based biosensors to meet the needs of practical application.

With deeper understanding of AMPs and improvement of related technologies such as smart formulation strategies and advanced chemical synthesis protocols, it is believed that the development and application of novel AMP-based agents, even though laborious, has a bright prospect.

## 4. Conclusions and Prospect

Due to the dormant character of persisters, traditional antibiotics that highly depend on active metabolism fail to kill these cells. As one of the most promising strategies against persisters, AMPs have been in the spotlight in recent decades. Currently, known AMPs are, or are synthesized based on short peptides generated as part of the defense system of most if not all living organisms, from protist to human. With activities like membrane perturbation, intracellular perturbation of metabolism and immunomodulation, AMPs are highly recommended as “next-generation antibiotics” with broad-spectrum antimicrobial activities. Furthermore, AMPs and their synergy with other certain anti-pathogen methods also show high efficiency against highly recalcitrant pathogenic persisters. Certain AMPs and their producers are powerful candidates as food preservatives to combat potential pathogens. Studies on AMP-based biosensing have made great progress to identify the presence of foodborne pathogens and their persisters, but the precision and accuracy still need to be improved. In the future, the complete understanding of structural features is essential to design or modify AMPs with increased stability, potency, and sensitivity as well as reduced hemolysis to meet the requirements of therapeutic and (food) product use.

Foodborne pathogens have caused severe impact both in terms of health as well as financially. Unfortunately, the existence of persistent pathogens related to food contamination as well as recurrent and chronic diseases significantly exacerbate this phenomenon. Persisters are non-heritable phenotypic variants that universally exist among almost all species of bacteria. They can be heterogeneously generated under normal conditions that allow part of the bacterial population to be pre-prepared for future stress. Persisters can also be induced under stress conditions and revert to wild types after stress removal to maintain bacterial vitality of the population as a whole. Molecular physiological research into the mode of actions of persister formation and their behavior in infection are much needed to understand the mechanisms at the basis of persister-cell formation both in the food harvesting/manufacturing chain as well as upon human consumption.

## Figures and Tables

**Figure 1 ijms-21-08967-f001:**
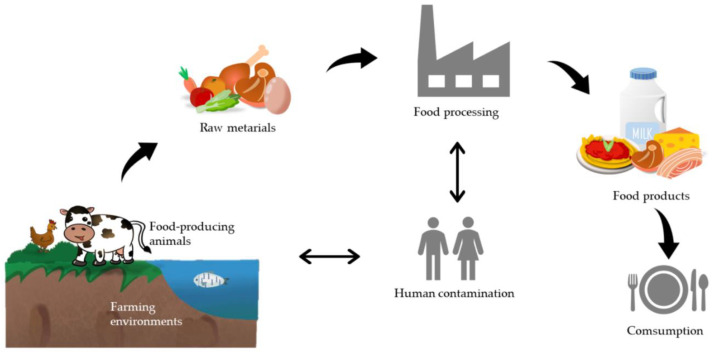
The possible sources of foodborne pathogens. Every step from farm to table can ultimately lead to foodborne diseases, including polluted farming environments, diseased food-producing animals, contaminated raw materials, non-hygienic food processing, and non-hygienic food consumption. In this process, food handlers can cause (post-processing) contamination, which they can also acquire themselves from animals or in the food processing environment even though the latter is less likely. The review focuses on persister-cells of various *Escherichia coli* (EPEC, EHEC, STEC, EIEC, EAEC, ETEC), *Salmonella*, and *Listeria* species both in food and upon human infection. For the major bacterial cause of gastroenteritis, *Campylobacter jejuni*, data on persister-cells is emerging but still scarce. *Staphylococcus aureus* and *Bacillus cereus* are considered human and animal contaminants of the food chain with the former being present on the skin of food handlers and the latter originating from spores present in the environment.

**Figure 2 ijms-21-08967-f002:**
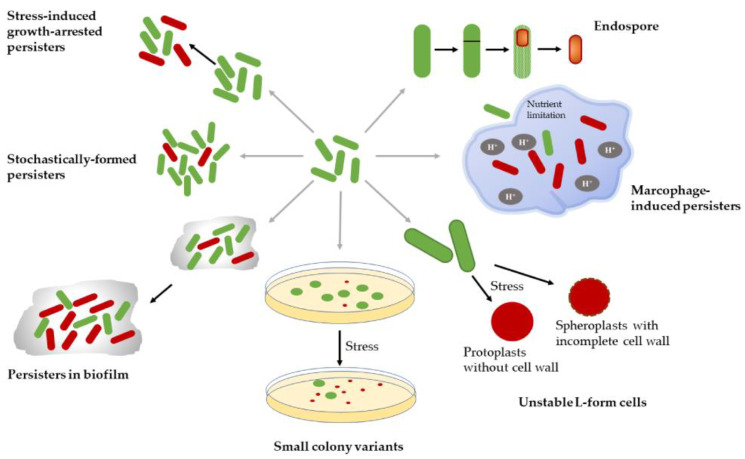
Different types of persister-cells and bacterial spores. Green images: susceptible bacteria cells. Red images: persisters. Images are not drawn to scale.

**Figure 3 ijms-21-08967-f003:**
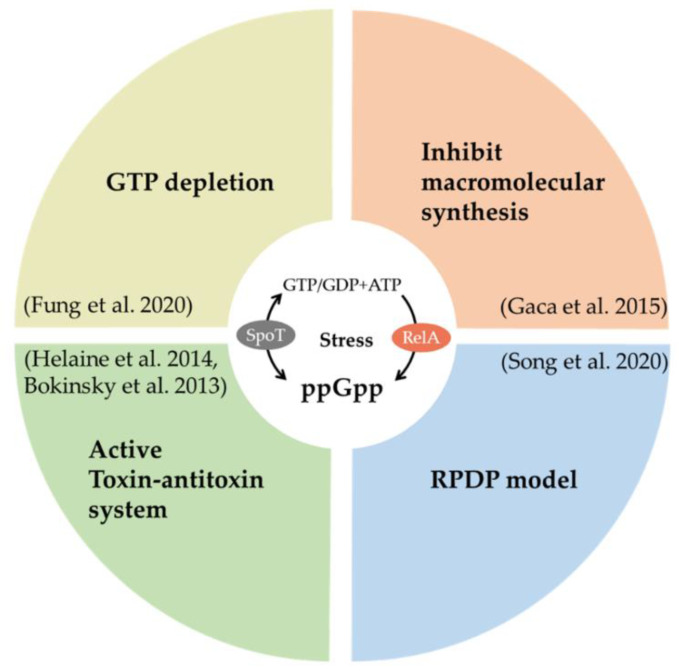
Functions of ppGpp associated with persister formation after being generated by a RelA/SpoT homolog family. RPDP model: ribosome dimerization persister model.

**Figure 4 ijms-21-08967-f004:**
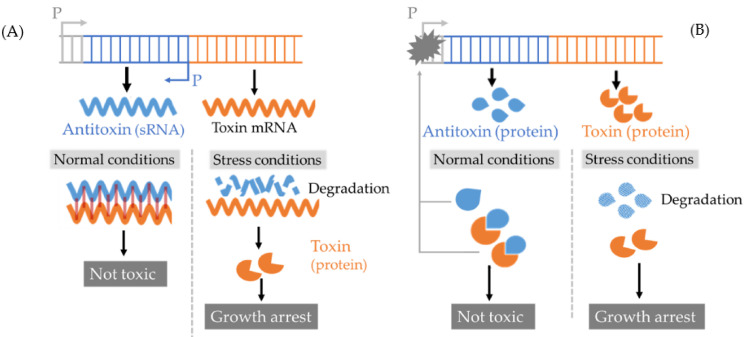
Models of Toxin-antitoxin (TA) systems associated with persister formation. (**A**) Type I TA system: it contains a small RNA (sRNA) antitoxin and a protein toxin. In normal conditions, antitoxin inhibits the translation of toxin mRNA. In a stress condition, the antitoxin is degraded, freeing the toxin mRNA to be translated. The type I toxin causes bacterial membrane depolarization. (**B**) Type II TA system: it contains a DNA-binding protein antitoxin and a protein toxin. In normal conditions, the antitoxin binds to the toxin and inhibits its activity. The antitoxin, as well as most of type II TA, the complex can target on the promoter and repress the transcription of the toxin gene. In stress conditions, the antitoxin is degraded by Lon or Clp proteases, releasing the toxin to inhibit several essential metabolic processes like DNA or protein synthesis. P: promoter.

**Figure 5 ijms-21-08967-f005:**
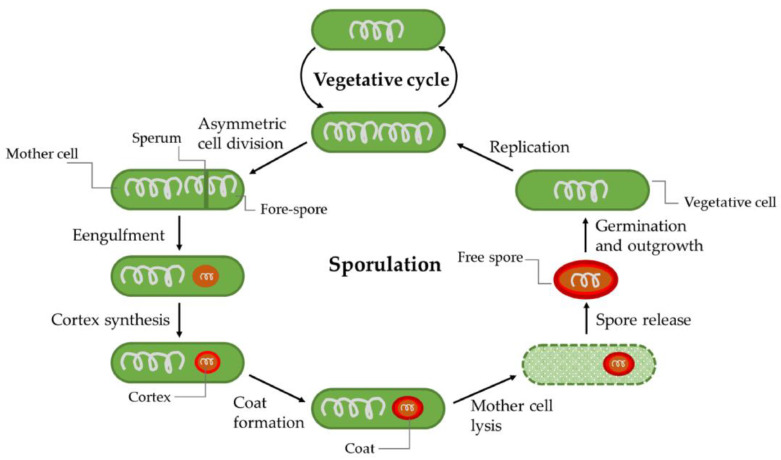
Model of the vegetative cycle and sporulation, germination, and outgrowth of *B. cereus*. Under severe stress conditions, *B. cereus* can initiate a sporulation process and form spores. Under favorable conditions, the spore can germinate and grow out to vegetative cells [56].

**Figure 6 ijms-21-08967-f006:**
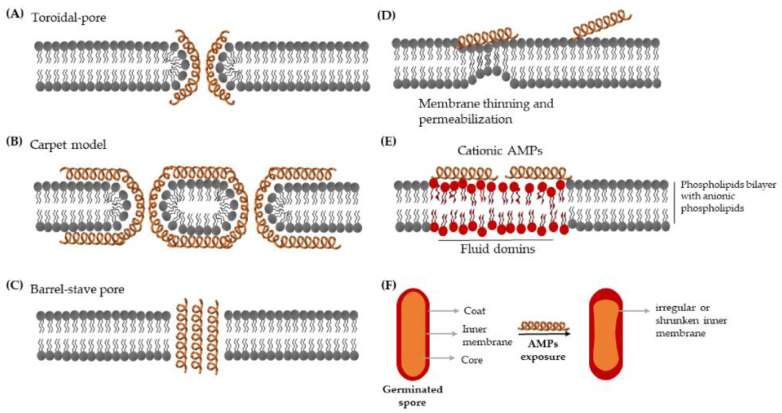
Various schematic models of antimicrobial peptides (AMPs) induced membrane disturbance. Models (**A**–**C**) are derived from in vitro membrane lipid model studies [123,124,125], and models (**D**–**F**) are based on single cell live-imaging studies [126,127,128]. See the manuscript text for a detailed explanation of each model.

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
