# Peer review of "Bacterial Persister-Cells and Spores in the Food Chain: Their Potential Inactivation by Antimicrobial Peptides (AMPs)"

_ijms, 2020, doi:10.3390/ijms21238967_

Round 1

Reviewer 1 Report

This is an interesting review on persisters and antimicrobial peptides. The presentation of persisters is comprehensive and may be a bit too dominant part of the review. The antimicrobial peptide part is not comprehensive, but it is an updated version of recent articles.

Title: AMP should be written in full.

Minor points:

Line 18: allows; change to: allow (plural)

  1. 21: has; change to: have (plural)
  2. 40: scientist, change to: scientists
  3. 80: an anti-persister agents; change to: an anti-persister agent
  4. 87: includes; change to: include (plural)
  5. 90: are; change to: is (singular)
  6. 216: inhibit; change to: inhibits (singular)
  7. 350: S.Typhimurium; change to: S. typhimurium
  8. 368+369: such B. cereus; change to: such as B. cereus

L 400: envelop; change to: envelope

L 560: has; change to: have (plural)

L 564: exacerbate; change to: exacerbates (singular)

Reviewer 2 Report

In this paper authors gave an overview of most common foodborne pathogens, their persisters and described possible applications of AMPs.

My remarks are:

  1. In line 58 and line 383 it is stated that AMPs are produced by bacteria and plants but it has to be written that AMPs can also be produced by human cells
  2. Section 2.1. Common foodborne pathogens is lengthily and authors should put focus only on the persisters of most common foodborne pathogens
  3. Mechanisms of modulation of host immune system by AMPs should be better described
  4. Specificity of AMPs should be discussed in section 3.3.
  5. Parameters that can influence action of some AMPs should be described (e.g. tissue localization and growth phase of the bacteria)

Reviewer 3 Report

There are a number of major issues with this review article that need to be addressed before it is ready for publication. The number one issue with the manuscript is that it reads as those a bunch of single sentence facts were put together without the thought of how they related to each paragraph, and then how each paragraph relates to the overall theme of the manuscript. It also jumps around a lot with now overall flow of thought about what the read is supposed to take home as a message. It reads as though first section is the pathogens, then we sort of have some facts about persister cell formation in those pathogens, and then we are discussing AMPs and it is kind of mentioned how that might have a role with persister (although not really). It is not clear the point or purpose of this manuscript, what is the “story” the authors are trying to tell with this information, instead it is just random facts with the foodborne pathogens as kind of the linking point.  

In addition, there are numerous grammatical errors that make the manuscript difficult to read in certain parts, which is amplified by the lack of a central theme or thought for paragraphs and overall manuscript. Please see below for specific examples.

Additionally, the foodborne pathogens that are discussed in the manuscript seem to be a little random compared to the information being presented, although again due to the lack of a central purpose to the paper it is difficult to be sure. From a foodborne disease perspective, Staphylococcus aureus is an intoxication therefore it will never be treated with antibiotics or AMPs as that would be pointless. Moreover, S. aureus contamination of food is due to contamination during preparation from improper hygiene of the food workers, therefore it does not persist in food processing facilities like some of the other pathogens. Thus, it is unclear why S. aureus is included in this manuscript as its persister cell role and AMPs are not important from a food safety prospective. The same can be said about Bacillus cereus, although it can have a role as a spore contaminate in food processing. Furthermore, spores are not persister cells and are a completely different physiological process and that should not be included in this manuscript. S. aureus and B. cereus should both be eliminated from this manuscript if the central theme is foodborne pathogens, persister cells and treatment with AMPs. Listeria monocytogenes is also mentioned in heavily in the beginning of the manuscript, but is barely described in the persister and AMPs sections of the manuscript. Why have it in the manuscript if there is not enough information to discuss it in detail for all the themes of the manuscript? The one time it is discussed in the persister cell section it is glossed over. Or at least mention why you discuss L. monocytogenes  so little (i.e. limited information, etc.). Finally, why is Campylobacter not discussed in the manuscript? Globally, it is the leading bacterial cause of gastroenteritis and seems like a critical foodborne pathogen to discuss in this manuscript as has been shown to form persister cells when exposed to ciprofloxacin (a known treatment) and is an effective biofilm former.

As previously mentioned, there are numerous grammatical and writing errors that need to be corrected, and it would be recommended to have several additional people proofread the manuscript. Although not a complete list, specifically here are some examples:

Line 29 – 31 – This sentence does not make sense, and should be corrected.

Line 70 – change “derivatives” to “its derivatives”

Line 80 – agents to agent or remove the “an”

Line 87 – “more than 99% serotypes” does not make sense

Line 91 – “which” to “that”

Line 94 – italicize Salmonella

Line 94 – “Salmonella cytolethal distending toxin” statement makes it sound like that is the only virulence factor needed for S. Typhi to produce disease and should be clarified.

Line 96 – Line 106 – Many Shiga toxin-producing E. coli (STEC) are not classified EHECs, while not all EHECs are associated with HUS, this should be clarified in this paragraph.

Line 115 – “joints” to “joint”

Line 126 – “fresh produces are” to “fresh produce has also been found to be a contributor”

Line 132 – “in the food matrix”

Line 138 – sentence has very weird phrasing and should be altered

Line 155 – “researches” to “research”

Line 158 – “secret” to “secrete”

Line 162 – “bacteria” to “bacterial”

Line 164 – an antibiotic

Line 206 – 208 – This sentence does not make sense and should be corrected.

Line 249 – “E. coli is the main species to study” does not make sense.

Line 256 – 260 – Not sure the point of this statement.

Line 273 – 274 – Just seems to end the paragraph no transition.

Line 291 – “$450 million costs per year” to “and costs nearly $450 million per year”

Line 294 – 297 – Seems to jump from human disease to industrial control without a transition

Line 306 – This sentence does not make sense, and should be corrected.

Line 311 – SCVs was used previously with this definition, should be used to first time it is used

Line 316 - This sentence does not make sense, and should be corrected.

Line 319 – 325 – Not sure the point of this paragraph, feels just throwing in some facts.

Line 328 – 333 – Need to re-write section.

Line 335 – Need better transition sentence for paragraph.

Line 346 – “an S. aureus” to “a S. aureus

Line 359 - This sentence does not make sense, and should be corrected.

Section 2.3.7 – Sporulation and Figure 5 should be eliminated

Figure 6 – Section B and C are different then how they are discussed in the text.

Line 447 – “immunity” to immune

Line 452 – 456 – Seems out of place and does not fit.

Line 459 – 460 – Not all bacteriocins are produced by lactic acid bacteria, this should be clarified.

Line 475 - This sentence does not make sense, and should be corrected.

Line 488 - This sentence does not make sense, and should be corrected.

Line 493 – 506 – This section seems out of place, no real flow, just putting sections in the manuscript. Overall flow should be corrected.

Line 519 – 522 - This sentence does not make sense, and should be corrected.

Line 539 – 541 - This sentence does not make sense, and should be corrected.

Line 543 – 551 – Unclear the reason for this section, is the suggestion AMP based treatment for foodborne infections? What about persister cell formation in food production/processing facilities?

Line 559 – 560 - This sentence does not make sense, and should be corrected.

Line 562 - This sentence does not make sense, and should be corrected.

Line 569 – 570 - This sentence does not make sense, and should be corrected.

The potential link between AMP and persister cell formation/treatment is obvious, however this manuscript fails to effective link them together.

Round 2

Reviewer 2 Report

Authors have removed most of the shortcomings of the paper with only one left:
In line 653 "Natural AMPs are ubiquitously synthesized by microbes, plants and animals...." humans have to be added as possible AMPs producers..

Author Response

Dear reviewer,

we thank you for your positive views on our revised manuscript and have made the addition required in line 420 of the revised text. The phrase highlighted by you was situated there and not at line 653!

kind greetings,

                   Stanley Brul

(corresponding author)